# TCM-Ladder: A Benchmark for Multimodal Question Answering on Traditional Chinese Medicine

**Jiacheng Xie**[1,2] **Yang Yu**[1,2] **Ziyang Zhang**[3] **Shuai Zeng**[1,2] **Jiaxuan He**[4]
**Ayush Vasireddy**[5] **Xiaoting Tang**[6] **Congyu Guo**[1,2] **Lening Zhao**[7] **Congcong Jing**[8]
**Guanghui An**[9*] **Dong Xu**[1,2*]

[1]Department of Electrical Engineering and Computer Science, University of Missouri, Columbia, MO, USA

[2]Christopher S. Bond Life Sciences Center, University of Missouri, Columbia, MO, USA

[3]Department of Computer Science, Northwestern University, Evanston, IL, USA

[4]Department of Computer Science and Mathematics, Truman State University, Kirksville, MO, USA

[5]Marquette High School, Chesterfield, MO, USA

[6]Community Health Service Center, Shanghai Pudong New Area, Shanghai, China

[7]School of Engineering and Applied Science, University of Pennsylvania, Philadelphia, PA, USA

[8]Department of Endocrinology, Seventh People's Hospital of Shanghai University of Traditional Chinese Medicine, Shanghai, China

[9]School of Acupuncture-Moxibustion and Tuina, Shanghai University of Traditional Chinese Medicine, Shanghai, China

*Corresponding authors

## Abstract

Traditional Chinese Medicine (TCM), as an effective alternative medicine, has been receiving increasing attention. In recent years, the rapid development of large language models (LLMs) tailored for TCM has highlighted the urgent need for an objective and comprehensive evaluation framework to assess their performance on real-world tasks. However, existing evaluation datasets are limited in scope and primarily text-based, lacking a unified and standardized multimodal question-answering (QA) benchmark. To address this issue, we introduced *TCM-Ladder*, the first comprehensive multimodal QA dataset specifically designed for evaluating large TCM language models. The dataset covers multiple core disciplines of TCM, including fundamental theory, diagnostics, herbal formulas, internal medicine, surgery, pharmacognosy, and pediatrics. In addition to textual content, TCM-Ladder incorporates various modalities such as images and videos. The dataset was constructed using a combination of automated and manual filtering processes and comprises over 52,000 questions. These questions include single-choice, multiple-choice, fill-in-the-blank, diagnostic dialogue, and visual comprehension tasks. We trained a reasoning model on TCM-Ladder and conducted comparative experiments against nine state-of-the-art general-domain and five leading TCM-specific LLMs to evaluate their performance on the dataset. Moreover, we proposed *Ladder-Score*, an evaluation method specifically designed for TCM question answering that effectively assesses answer quality in terms of terminology usage and semantic expression. To the best of our knowledge, this is the first work to systematically evaluate mainstream general-domain and TCM-specific LLMs on a unified multimodal benchmark. The datasets and leaderboard are publicly available at `https://tcmladder.com` and will be continuously updated. The source code is available at `https://github.com/orangeshushu/TCM-Ladder`.

39th Conference on Neural Information Processing Systems (NeurIPS 2025) Track on Datasets and Benchmarks.

# 1 Introduction

The development of large language models (LLMs) tailored to the field of Traditional Chinese Medicine (TCM) [1, 2] has emerged as a significant research direction. Given the unique and intricate nature of the TCM knowledge system, the construction of intelligent tools specifically designed for this domain can substantially improve the efficiency of medical students, clinicians, and researchers. Such models have the potential to facilitate accurate and timely access to specialized information for clinical decision-making, knowledge retrieval, and academic inquiry, thereby supporting effective reasoning and practical application within the TCM framework.

TCM diagnostic methods including inspection, auscultation and olfaction, inquiry, and palpation embody a representative process of multimodal information acquisition, integration, and reasoning [3]. Fundamentally, this diagnostic paradigm reflects the nature of multimodal fusion in clinical decision-making. However, existing LLMs tailored for TCM still face notable limitations in real-world applications. These limitations are primarily manifested in their relatively small model scales, insufficient reasoning capacity, and the lack of deep integration of multimodal information. The acquisition of high-quality TCM data poses significant challenges, as it requires deep expertise in traditional medicine, sustained clinical data collection, and extensive manual annotation. Currently, most mainstream medical benchmark datasets [4, 5, 6, 7, 8] are predominantly focused on Western medicine and have yet to systematically address the core tasks unique to TCM, including syndrome differentiation, symptom-based diagnosis, and formula-herb matching. Furthermore, the training and evaluation of existing TCM large language models remain heavily reliant on unimodal textual data, neglecting other essential modalities that are widely utilized in clinical practice. These include diagnostic images (e.g., tongue and pulse), medicinal herb atlases, and structured case records. Such an overdependence on textual data severely constrains the models' ability to capture the holistic and multimodal nature of TCM knowledge, thereby impeding their performance in complex and real-world clinical scenarios.

Therefore, the construction of a standardized evaluation dataset for TCM that integrates text, images, audio, and structured data is of great importance. On one hand, such a dataset would enable a comprehensive and accurate assessment of existing LLMs in handling complex multimodal tasks, thereby providing a realistic reflection of their overall performance in clinical applications. On the other hand, a unified and standardized evaluation framework would facilitate fair and objective comparisons across different TCM-specific models, supporting continuous optimization and iterative improvement of model capabilities.

To address the aforementioned gaps, we proposed *TCM-Ladder*, which, to the best of our knowledge, is the first large-scale multimodal dataset specifically designed for the training and evaluation of large language models in TCM. TCM-Ladder encompasses a wide spectrum of domain-specific knowledge, including fundamental TCM theories, diagnostics, formulae, pharmacognosy, clinical medicine, as well as visual modalities such as tongue images, herbal medicine illustrations, acupuncture, and tuina (therapeutic massage), thereby offering a comprehensive foundation for developing and benchmarking TCM-specific LLMs.

As illustrated in Figure 1, we designed a series of evaluation tasks based on the TCM-Ladder dataset to comprehensively evaluate the capabilities of TCM-specific LLMs across multiple dimensions. We constructed a total of 21,326 high-quality questions and 25,163 diagnostic long-text dialogues based on domain-specific literature and publicly available databases across various subfields of TCM. In addition, we released a visual dataset comprising 6,061 images of medicinal herbs, 1,394 tongue images, 6,420 audio clips, and 49 videos, forming a comprehensive multimodal foundation to support diverse evaluation tasks. All textual and visual data were independently reviewed and validated by certified TCM practitioners to ensure accuracy, clinical relevance, and authoritative quality. Subsequently, we benchmarked the performance of nine state-of-the-art general-domain LLMs [9, 10, 11, 12, 13, 14, 15, 16, 17] and five TCM-specific models [18, 19, 20] using the TCM-Ladder dataset. Additionally, we fine-tuned a GPT-4-based model, *BenCao* [21, 22], and trained a Qwen2.5-7B based reasoning model [23, 24], which uses a training subset constructed from TCM-Ladder to support TCM-specific reasoning tasks.

Our contributions can be summarized as follows:

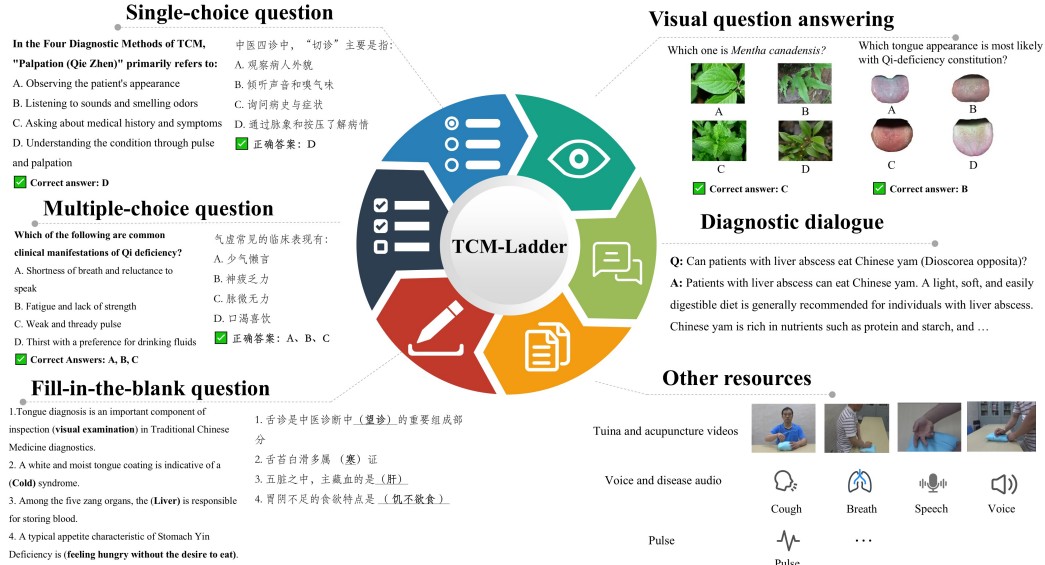

Figure 1: Overview of the architectural composition of TCM-Ladder. TCM-Ladder encompasses six task types aimed at evaluating the comprehensive capabilities of large language models in Traditional Chinese Medicine. These include: (1) single-choice questions, which assess basic knowledge recognition; (2) multiple-choice questions, designed to test the model's ability to integrate and reason over complex concepts; (3) long-form diagnostic question answering, which evaluates clinical reasoning based on detailed symptom descriptions and patient inquiries; (4) fill-in-the-blank tasks, which measure generative accuracy and contextual understanding without the aid of answer options; (5) image-based comprehension tasks, involving the interpretation of medicinal herb and tongue images to assess multimodal reasoning across visual and textual inputs; and (6) additional audio and video resources, such as diagnostic sounds, pulse recordings, and tuina (massage) videos, which support the development and evaluation of multimodal TCM models incorporating auditory and dynamic visual data.

- We constructed TCM-Ladder, a multimodal dataset designed for both training and evaluating TCM-specific and general-domain LLMs. The dataset encompasses multiple TCM sub-disciplines and multiple data modalities.

- We designed a comprehensive set of tasks including single-choice questions, multiple-choice questions, fill-in-the-blank, visual understanding tasks, and long-form question answering to evaluate models' reasoning and comprehension abilities across diverse tasks.

- We introduced *Ladder-Score*, an evaluation metric that integrates TCM-specific terminology and LLM-assisted semantic scoring to assess terminological accuracy and reasoning quality in TCM question answering.

- We systematically evaluate the performance of nine general-domain and five TCM-specific LLMs on TCM-Ladder. To the best of our knowledge, this is the first work to conduct a comparative evaluation of diverse LLMs on a unified multimodal TCM dataset.

- We developed an interactive data visualization website that not only presents evaluation results but also allows researchers to explore existing data and contribute new entries, thereby providing a standardized, extensible, and multimodal infrastructure for future benchmarking of TCM-specific LLMs.

## 2 Related Works

In recent years, the expanding application of LLMs in medicine and the biomedical sciences has driven the progressive development of evaluation datasets tailored for TCM, evolving from modern medical domains to TCM-specific tasks, and from classification-based to generation-based paradigms.

As shown in Table 1, Huatuo-26M [25], released in 2020, remains the largest Chinese medical question-answering (QA) dataset, comprising over 26 million question–answer pairs sourced from online encyclopedias, medical knowledge bases, and telemedicine transcripts. Despite its scale, the dataset is affected by noisy labels, informal expressions, redundancy, and a lack of TCM-specific annotations, limiting its utility for TCM applications. CBLUE [26] introduced a standardized multi-task evaluation suite for Chinese biomedical natural language processing (NLP), covering tasks such as named entity recognition, relation extraction. PromptCBLUE [27] extended this framework via instruction tuning and prompt reformulation to facilitate few-shot and zero-shot evaluation. However, both benchmarks were designed around modern medical reasoning and do not capture the unique logic or semantic structure of TCM diagnosis.

To address these gaps, TCMBench [28] compiled 5,473 structured questions from national TCM licensing examinations, providing a focused benchmark for foundational knowledge assessment. Nevertheless, it lacks multimodal input (e.g., tongue and pulse images) and real-world diagnostic reasoning tasks. TCMEval-SDT [29] introduced syndrome differentiation based on 300 clinical cases, evaluating the model's reasoning over symptom–pathomechanism–syndrome chains. While it improved interpretability, its scale and disease diversity remained limited. Subsequently, TCM-3CEval [30] proposed a cognitive three-axis framework, including basic knowledge, classical text comprehension, and clinical decision-making, enabling fine-grained cognitive evaluation. However, tasks were still text-only and often reduced the complexity of classical TCM literature to overly simplistic answers. TCMD [31] presented a human-annotated open-ended QA benchmark that emphasizes reasoning and generation, although annotation costs limited its scale and case diversity. ShenNong_TCM_Dataset [32] adopted a novel approach, combining knowledge graphs with ChatGPT-based generation to create over 110,000 instruction–response pairs on herbal medicine and treatment plans. While valuable for instruction tuning, the absence of expert validation raises concerns over factual accuracy and stylistic fidelity. CHBench [33] introduced a safety-focused benchmark with 9,492 community-sourced questions, highlighting deficiencies in LLM reliability under ethically sensitive conditions. However, its scope remains narrow. MedBench [34] represents the most comprehensive Chinese medical LLMs evaluation to date, integrating 20 datasets and over 300,000 questions across diverse tasks, including QA, clinical case analysis, diagnostic reasoning, and summarization. The platform supports dynamic sampling and randomized option ordering to prevent overfitting. However, access to API use is restricted due to data privacy concerns. Benchmarks like CMB [35] and CMExam [36] further extend to structured exam QA, offering high coverage but lacking realistic patient–physician interaction.

Table 1: Overview of TCM and medical QA datasets. En: English, Zh: Chinese

| Dataset | Format | TCM Coverage | Size | Source | Domain | Task | Verified | Language |
|---|---|---|---|---|---|---|---|---|
| Huatuo-26M [25] | Text | ✗ | 26,000,000+ | Online QA platforms and physician records | Medicine | QA, Dialogue | ✗ | Zh |
| CBLUE [26] | Text | ✗ | 13 subtasks | Clinical trials, EHRs, logs, textbooks | Biomedical | Classification, NER, RE, NLI | Partial | Zh |
| PromptCBLUE [27] | Text | ✗ | 11 prompt datasets | Prompt-formatted CBLUE | Biomedical | Same as CBLUE | ✗ | Zh |
| TCMD [31] | Text | ✓ | 1,500+ | Professional TCM practitioners | TCM | NER, Term Normalization | ✓ | Zh |
| TCM-3CEval [30] | Text | ✓ | 4,000+ | Expert-annotated multi-rater QA | TCM | QA | ✓ | Zh |
| ShenNong_TCM_Dataset [32] | Text | ✓ | 113,000 | TCM knowledge graph, GPT-3.5 assisted | TCM | Dialogue | ✗ | Zh |
| CMB [35] | Text | Partial | 280,839 MCQ, 74 consults | Textbooks, forums, exams | TCM | MCQ, Dialogue | ✓ | Zh |
| CMExam [36] | Text | Partial | 60,000+ | TCM licensing exam | Medicine | MCQ, QA | Partial | Zh |
| CHBench [33] | Text | Partial | 9,492 | Community health Q&A | Health | QA | ✓ | Zh |
| MedBench [34] | Text | Partial | 40,041 | Clinical exam questions | Medicine | MCQ, QA | Partial | Zh |
| TCMBench [28] | Text | ✓ | 5,473 | TCM licensing exam | TCM | QA | ✗ | Zh |
| TCM-Ladder (Ours) | Text, images, audio, video | ✓ | 52,000+ | Research, books, exams, online medical QA platforms | TCM | MCQ, FIB, QA, Dialogue, Image Understanding | ✓ | Zh & En |

TCM-Ladder distinguishes itself from existing datasets in several key aspects. First, it establishes a large-scale, open-ended QA dataset that spans a wide range of TCM subfields, including fundamental theory, diagnostics, herbal formulas, internal medicine, surgery, pharmacognosy and pediatrics. This breadth enables more thorough and representative evaluation of TCM-specific LLMs across multiple knowledge domains. Second, TCM-Ladder incorporates visual elements, including herbal medicine images and tongue diagnostics. This multimodal design reflects TCM diagnostic practices, requiring LLMs to demonstrate both textual reasoning and visual understanding capabilities. Third, TCM-Ladder incorporates a variety of task formats. This comprehensive task structure facilitates

an in-depth evaluation of the strengths and limitations of LLMs, providing guidance for the future development of TCM-specific models.

## 3 TCM-Ladder Dataset

### 3.1 Data Collection

We collected a question-answering dataset covering various domains of TCM, including several publicly available datasets previously published in academic literature under permissive licenses. For the textual data, we identified seven subfields: fundamental theory, diagnostics, herbal formulas, internal medicine, surgery, pharmacognosy, and pediatrics.

Regarding herbal medicine images, we collected over 6,061 images of medicinal herbs based on the names referenced in the *Pharmacology of Chinese Herbs* [37]. The dataset comprises images sourced from publicly available online resources, as well as photographs we captured at traditional Chinese medicine manufacturing facilities. Sample images and the collection process are provided in **Appendix G**.

The clinical tongue images were collected by a tongue imaging device [14] at Shanghai University of Traditional Chinese Medicine. This device is designed for tongue diagnosis and provides stable and consistent lighting conditions during image acquisition. Another subset of the proprietary data was obtained from our previous work, the *iTongue* [38, 39] diagnostic software. All data collection procedures were approved by the institutional ethics review board. To protect the privacy of tongue image contributors, only a subset of tongue image patches and corresponding labels has been released.

The video data was recorded by faculty members from the Department of Acupuncture-Moxibustion and Tuina at Shanghai University of Traditional Chinese Medicine. These instructional videos cover essential techniques, procedural explanations, and key operational steps. Audio and pulse diagnosis data were sourced from publicly available datasets referenced in academic publications [40, 41, 42, 43]. We manually filtered and removed samples with poor quality or missing information from the collected data.

### 3.2 Construction of the Datasets

The textual QA data consisted of two parts. The first part comprises 5,000 TCM-related QA pairs manually written by licensed TCM practitioners following a standardized question design protocol (see **Appendix I**). To ensure answer accuracy, each question was independently reviewed and verified by two additional TCM physicians. The second part of the textual QA data was collected from publicly available sources, including the *National Physician Qualification Examination of China* and various open-access online resources. Detailed data sources and construction guidelines are provided in **Appendix B**.

The visual question-answering (VQA) tasks were constructed through both manual annotation and automated generation based on existing knowledge bases. For the manually created subset, domain experts selected high-quality images from the herbal medicine image repository and generated corresponding questions based on each herb's name and medicinal properties. The automatically generated subset was produced through a procedural pipeline. For example, an image labeled as *Astragalus membranaceus* (Huangqi) was selected as the correct answer, while three distractor images were randomly sampled from the knowledge base. A question was then constructed using a predefined template library, such as "Which of the following images shows *Huangqi*?" The design of tongue image understanding tasks followed a similar approach. Details of the construction process and implementation code can be found in **Appendix G**.

### 3.3 Deduplication and Preprocessing

Detecting duplication and semantic similarity in the data is critical for both model evaluation and training, as it helps prevent evaluation failures and reduces the risk of overfitting caused by redundant content. Given the diverse sources of the original data, we conducted a comprehensive similarity detection process on the aggregated dataset and removed highly similar questions to enhance overall data quality. The methods employed included string edit distance [44], TF-IDF [45, 46] with cosine similarity, and BERT-based [47, 48] semantic encoding. Subsequently, all questions and answers

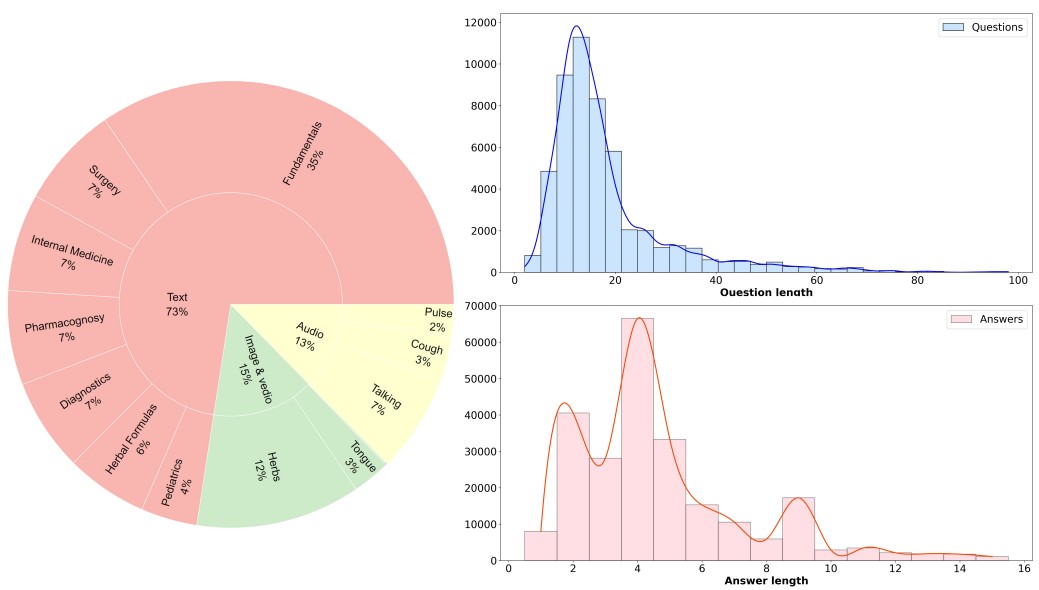

Figure 2: Data distribution and length statistics in TCM-Ladder. The left illustrates the dataset composition across text, image, and audio modalities, along with TCM subfields. The right plots show the distribution of question and answer lengths.

were manually reviewed by two licensed physicians. The selection criteria and detailed experimental procedures are provided in **Appendix I**. Subsequently, we divided the dataset into three subsets: 10% for evaluation, 10% for validation, and 80% for training. To ensure balanced representation, each subset contains question-answer pairs spanning all subfields.

### 3.4  Datasets Statistics

Table 2 presents the statistics of all constructed question-answer pairs across different categories. The TCM-Ladder dataset comprises 52,169 TCM-related QA instances, including 6,061 herbal medicine images and 1,394 annotated tongue image patches. The distribution of each data type is illustrated in Figure 2.

## 4  Ladder-Score

Evaluating free-form question answering presents notable challenges, as the responses are often descriptive and lack a predefined standard format. This issue is further exacerbated in the context of TCM diagnostic tasks, where large language models are capable of generating diverse and nuanced answers. Even when the expressions differ, the underlying responses may still be factually correct. Traditional evaluation metrics such as BLEU [49] and ROUGE [50] often fail to capture this semantic equivalence adequately. Recently proposed methods [51, 52, 53] employ instruction-tuned models to score candidate answers on a rubric-based scale. We propose a novel evaluation metric for TCM question answering, named Ladder-Score. This score comprises two components: *TermScore*, which assesses the accuracy and completeness

Table 2: Statistics of the collected questions

| Statistics | Number |
| --- | --- |
| Total questions | 52,169 |
| Total answers | 238,867 |
| Total subjects | 7 |
| Maximum question length | 98 |
| Maximum answer length | 16 |
| Average question length | 18 |
| Average answer length | 5 |
| Total images | 7,455 |
| Herbs visual questions | 6,061 |
| Tongue visual questions | 1,394 |
| Total videos | 49 |
| Total audios | 6,420 |

of TCM terminology usage, and *SemanticScore*, derived from LLMs to evaluate multiple aspects including logical consistency, semantic accuracy, comprehensiveness of knowledge, and fluency of expression. As shown in Equation (1), the Ladder-Score is a weighted combination of these two components:

$$\text{Ladder-Score} = \alpha \cdot \text{TermScore} + \beta \cdot \text{SemanticScore} \qquad (1)$$

where $\alpha = 0.4$ and $\beta = 0.6$,which can be adjusted based on practical needs. The scoring criteria, terminology dictionary, and calculation examples can be found in **Appendix H**.

## 5 Experiments

### 5.1 Experiment Setup

We evaluated nine state-of-the-art general-domain LLMs and five TCM-specific models on the TCM-Ladder dataset across five task settings: single-choice questions, multiple-choice questions, fill-in-the-blank questions, image-based understanding, and long-form dialogue tasks. Evaluations were conducted under zero-shot settings, and models received only the task instructions as input. For single-choice and image understanding tasks, we used the top-1 prediction accuracy [54] as the primary evaluation metric. For multiple-choice tasks, we adopted exact match accuracy to assess performance comprehensively. For fill-in-the-blank and long-form dialogue tasks, we evaluated models using metrics such as accuracy, BLEU [49], ROUGE [50], METEOR [55] and BERTScore [56]. The detailed evaluation environment can be found in **Appendix D**.

### 5.2 Model Training

We trained two models using the TCM-Ladder dataset. The first is BenCao [21], an online model fine-tuned from ChatGPT, and the second is *Ladder-base*, which is built upon the pretrained Qwen2.5-7B-Instruct [57] model and enhanced with Group Relative Policy Optimization (GRPO) [58] to improve its reasoning capabilities. The BenCao model was trained on knowledge extracted from over 700 classical Chinese medicine books, none of which contained any question-answer pairs. Additionally, the training subset of TCM-Ladder was used as its knowledge base.

The GRPO stage for Ladder-base was conducted on two NVIDIA A100 PCIe GPUs (80GB each). The temperature and top-p sampling of Ladder-base were 0.7 and 0.8. Training was performed for 2 epochs with a group size of 6 and a batch size of 12, resulting in a total training time of approximately 60 hours. Model training and inference were implemented using HuggingFace Transformers, while the GRPO process was carried out using the TRL (Transformer Reinforcement Learning) library [59]. Details of the training process can be found in **Appendix C**.

### 5.3 Human Evaluation

We conducted a human evaluation using 20% of the TCM-Ladder test set. Due to the coverage of multiple subfields, establishing a reliable human upper bound poses a significant challenge, as accurately answering questions across all domains requires extensive interdisciplinary expertise. To investigate this issue, we recruited two licensed clinical TCM physicians, both holding senior titles and not involved in the original data annotation. Human evaluators were asked to select the correct answers based on the question stems and to identify the correct herbal medicine and tongue images. During the evaluation process, both physicians emphasized the challenge of maintaining high confidence across all domains. For example, although they are highly knowledgeable about the pharmacological properties and clinical applications of herbal medicines, they encountered difficulties when asked to identify herbs solely based on images. The challenge became especially evident when the herbs appeared in different visual forms, such as raw botanical specimens, dried slices, or moist decoctions, which often vary significantly in appearance. According to their feedback, such recognition tasks, especially those involving distinctions among various processed forms of herbs, are better handled by trained dispensary pharmacists than by clinical practitioners. In terms of top-1 accuracy for answer retrieval, the human evaluators achieved a performance of 64%, which was approximately 4% lower than that of the best-performing model (BenCao). This suggests that

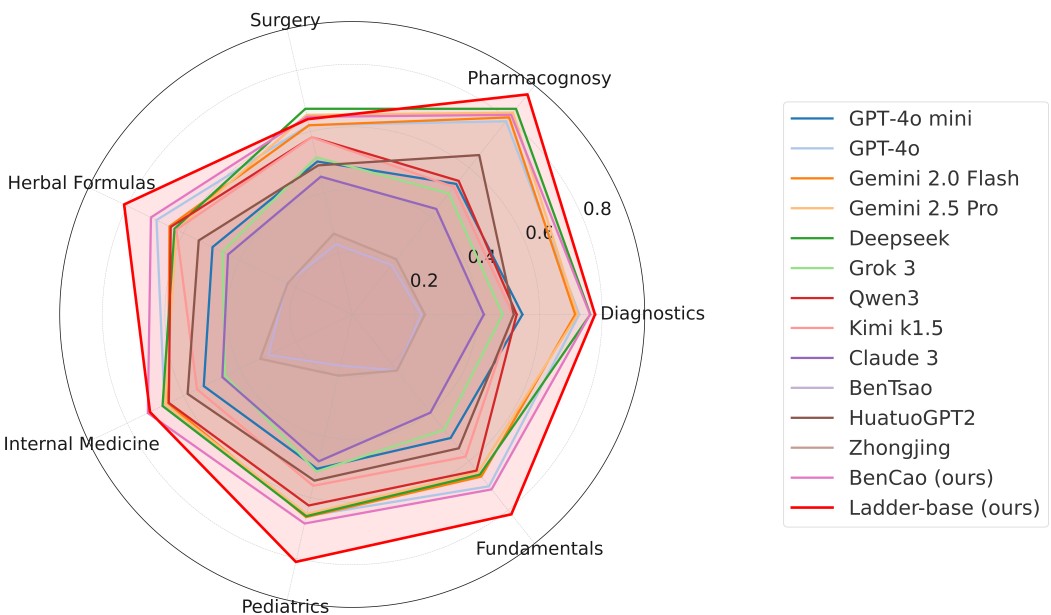

Figure 3: Performance of general-domain and TCM-specific language models on single and multiple-choice question answering tasks.

LLMs may already possess strong comprehension capabilities in the domains of herbal medicine and tongue image recognition.

## 5.4 Main Results

### 5.4.1 Text-Based Single and Multiple-Choice Question Answering

As shown in Figure 3, Ladder-base consistently outperforms other models across all subject areas, achieving the highest overall accuracy. Notably, its performance is especially strong in Pharmacognosy, Herbal Formulas, and Pediatrics, where exact match scores exceed 0.85. Our model, BenCao, also demonstrates robust performance, particularly in Diagnostics and Internal Medicine. Among the general-domain LLMs, Gemini 2.5 Pro, Deepseek, and Qwen3 show relatively stable accuracy across domains, with scores ranging from 0.65 to 0.75, though they still fall short compared to domain-specific models. In contrast, Claude 3, GPT-4o mini, and BenTsao underperform, especially in the more clinically nuanced domains such as Surgery and Pediatrics, suggesting limited capability in handling complex, multi-faceted TCM tasks. These findings highlight the advantage of domain-specific fine-tuning and multi-source integration, as utilized in Ladder-base, for enhancing the accuracy and generalization of LLMs on structured TCM knowledge assessments.

### 5.4.2 Visual Question Answering

To further assess the models' capability in visual understanding tasks within TCM, we evaluated ten LLMs on two image-based benchmarks: herbs classification and tongue image diagnosis. As illustrated in Figure 4, performance varies considerably across models. Among the evaluated models, BenCao achieves the highest accuracy in both tasks, with over 80% on herb recognition and above 65% on tongue classification, demonstrating strong multimodal understanding grounded in TCM-specific training. General-domain LLMs such as Gemini 2.5 Pro, Gemini 2.0 Flash, and Qwen3 exhibit moderate performance, with herb classification accuracy around 65–75%, but show a relative drop in tongue image tasks (around 50–60%), likely due to the greater complexity and domain specificity of tongue diagnosis.

In contrast, models like GPT-4o, Claude 3, Kimi k1.5, and Grok 3 demonstrate limited performance, particularly in the tongue classification task, where accuracies fall below 40%. This reveals their insufficient visual comprehension of TCM-related imagery. Notably, models such as Ladder-base

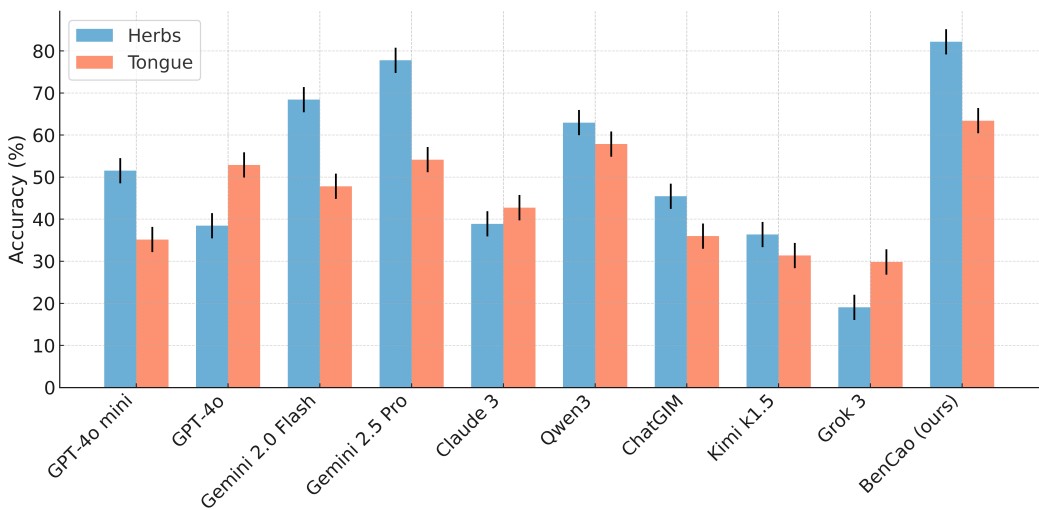

Figure 4: The performance of large language models on questions regarding Chinese herbal medicine and tongue image classification.

and Zhongjing are excluded from this figure because their current architectures do not support image understanding. Their current design focuses on structured text-based TCM evaluation and does not support visual input.

### 5.4.3 Diagnostic Dialogue and Fill-in-the-Blank Questions

As shown in Table 3, in the diagnostic dialogue task, our model Ladder-base achieved the highest scores in BLEU-4 (0.0249), and ROUGE-L (0.2431), while also maintaining a strong Ladder-Score (0.803). This indicates that Ladder-base generates answers with high lexical similarity, semantic accuracy, and alignment with TCM diagnostic logic. Notably, Qwen3 achieved the best Ladder-Score (0.861) and the highest METEOR (0.2328), showcasing its strength in generating fluently worded responses. BenCao achieved the best BERTScore (0.9663), reflecting its semantic closeness to gold references.

In the fill-in-the-blank task, BenCao significantly outperformed all other models, achieving the highest exact match accuracy of 0.9034, followed by Qwen3 (0.8786) and Deepseek (0.874). Our Ladder-base model also performed competitively with 0.8623 accuracy, further demonstrating its generalizability beyond free-form dialogue. Overall, the results demonstrate that Ladder-base excels in structured diagnostic dialogue tasks, generating semantically accurate and logically coherent responses, while BenCao shows outstanding performance in fill-in-the-blank tasks, reflecting strong factual recall and precise terminology usage. Domain-specific models consistently outperform general-domain LLMs, particularly in tasks that require accurate retrieval of structured TCM knowledge and professional terms.

## 6 Application Website

In addition to releasing the raw dataset, we provide access to all TCM-Ladder data and leaderboard results through an interactive website (`https://tcmladder.com/`). This platform enables researchers to explore, verify, and contribute to the open-access data. We encourage the research community to submit additional data through the platform, and we intend to expand the dataset continuously as part of our ongoing efforts. Our objective is to establish a long-term and reliable data foundation for the training and evaluation of TCM-specific LLMs.

Table 3: Performance comparison on diagnostic dialogue and fill-in-the-blank tasks

| Model | Diagnostic dialogue | | | | | Fill-in-the-blank |
|---|---|---|---|---|---|---|
| | BLEU-4 | ROUGE-L | METEOR | BERTScore | Ladder-Score | Exact match accuracy |
| GPT-4o mini | 0.0034 | 0.1125 | 0.1190 | 0.9433 | 0.718 | 0.4320 |
| GPT-4o | 0.0040 | 0.1447 | 0.2073 | 0.9620 | 0.828 | 0.5140 |
| Gemini 2.0 Flash | 0.0067 | 0.1518 | 0.2155 | 0.9633 | 0.836 | 0.4360 |
| Gemini 2.5 Pro | 0.0180 | 0.1353 | 0.2393 | 0.9605 | 0.859 | 0.7143 |
| Deepseek | 0.0047 | 0.1533 | 0.1293 | 0.9455 | 0.825 | 0.8740 |
| Grok 3 | 0.0063 | 0.1751 | 0.1691 | 0.9526 | 0.686 | 0.6389 |
| Qwen3 | 0.0225 | 0.1818 | **0.2328** | 0.9642 | **0.861** | 0.8786 |
| Kimi k1.5 | 0.0100 | 0.1878 | 0.1586 | 0.9559 | 0.708 | 0.8378 |
| Claude 3 | 0.0068 | 0.2267 | 0.2203 | 0.9561 | 0.756 | 0.4890 |
| BenTsao | 0.0024 | 0.1135 | 0.1725 | 0.9531 | 0.613 | 0.1620 |
| HuatuoGPT2 | 0.0086 | 0.1375 | 0.1742 | 0.9635 | 0.855 | 0.2347 |
| Zhongjing | 0.0044 | 0.1951 | 0.1134 | 0.9539 | 0.573 | 0.2167 |
| BenCao (ours) | 0.0073 | 0.2156 | 0.2013 | **0.9663** | 0.791 | **0.9034** |
| Ladder-base (ours) | **0.0249** | **0.2431** | 0.2268 | 0.9549 | 0.803 | 0.8623 |

# 7 Limitations and Societal Impact

Although TCM-Ladder encompasses question-answer pairs from multiple disciplines within TCM, its current scale remains insufficient to cover the full breadth of TCM knowledge. TCM diagnosis is inherently a multimodal process, in which textual information represents only one component. At present, the utilization of data related to tongue diagnosis, pulse diagnosis, and olfactory inspection remains limited, and these modalities require further supplementation and enrichment. Expanding and continuously updating the scope and scale of data included in TCM-Ladder will be a critical direction for future research. It is also important to acknowledge that the current dataset was primarily derived from Chinese clinical populations, which constrains demographic diversity, particularly in terms of ethnicity. Such geographical and cultural specificity may introduce bias when extrapolating findings to broader populations. Future extensions of TCM-Ladder will aim to incorporate more demographically and regionally diverse samples to improve fairness, inclusivity, and generalizability across different healthcare contexts. Additional discussions can be found in **Appendix J**.

# 8 Conclusion

We introduced TCM-Ladder, the first multimodal benchmark dataset designed explicitly for evaluating LLMs in the context of TCM. In addition, we proposed a novel evaluation metric, Ladder-Score, which enabled more precise analysis of the semantic alignment between candidate and reference answers. We conducted comprehensive experiments involving nine state-of-the-art general-domain and five TCM-specific LLMs, marking the first systematic comparison on a unified benchmark. Furthermore, we fine-tuned two open-source models using a subset of TCM-Ladder, and observed significant performance improvements over zero-shot baselines. Our work established a reproducible and extensible benchmark for TCM-specific, providing a foundation for future development and evaluation in this emerging research area.

## Acknowledgements

This work was supported by Paul K. and Diane Shumaker Endowment Fund at University of Missouri.

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
