# OpenReview forum: "TCM-Ladder: A Benchmark for Multimodal Question Answering on Traditional Chinese Medicine"
_NeurIPS.cc/2025/Datasets_and_Benchmarks_Track — NeurIPS 2025 Datasets and Benchmarks Track poster_

### Official Review · Reviewer_rBn6 · 2025-07-01

**Rating:** 4
**Confidence:** 4

**Summary:**

The paper introduces TCM-Ladder, a benchmark dataset for multimodal question answering in Traditional Chinese Medicine (TCM), which incorporates four types of modalities: text, images, audio, and video. It spans multiple TCM subfields, such as fundamental theory, diagnostics, and herbal formulas. The authors also propose a novel evaluation metric called Ladder-Score, which combines terminology accuracy with semantic consistency. They conduct systematic evaluations on several mainstream general-purpose language models and TCM-specific models. Additionally, the dataset, model training details, code, and evaluation platform have been made publicly available, facilitating reproducibility and extensibility for the research community. However, the current task design is heavily focused on structured formats, lacking generative and open-ended question answering tasks.

**Dataset Code Accessibility:**

Yes

**Dataset Code Comments:**

I answered "Yes" because the authors have made both the dataset and code publicly available, along with detailed documentation.

**Ethical Comments:**

I did not identify any significant ethical concerns with this work. The authors have addressed potential issues related to data privacy and human subjects by obtaining IRB approval for the collection of tongue images and ensuring that all personally identifiable information is anonymized. The dataset is carefully curated and verified by certified TCM practitioners, which reduces the risk of misinformation. Additionally, the open release of data and models is accompanied by safeguards and proper licensing. While the work involves applications in healthcare, it does not make clinical claims or promote direct deployment in medical settings without expert oversight, thereby minimizing potential societal harm.

**Ethical Considerations:**

No, there are no or only very minor ethics concerns

**Final Justification:**

The authors have responded to each of our concerns and provided a clear outlook for future work. However, their replies did not fully or satisfactorily address the issues we raised. That said, I acknowledge that their work fills an important gap in the field, and therefore I am still inclined to give a positive score to this submission.

**Limitations Weaknesses:**

The benchmark primarily focuses on closed-ended question answering, with relatively few generative or open-ended QA tasks.

Although the dataset includes audio and video modalities, the current evaluation tasks are primarily focused on text and image. The potential of modalities such as pulse diagnosis audio and acupuncture videos remains underexplored.

While the proposed metric is conceptually sound, it would be strengthened by ablation studies or alignment analysis with human evaluations to validate the individual contributions of TermScore and SemanticScore.

**Strengths Contributions:**

The contribution is clear and addresses a critical gap. The field of Traditional Chinese Medicine currently lacks a systematic benchmark for multimodal evaluation. TCM-Ladder takes the lead with its diverse task types, rich modality coverage, and broad domain scope.

The dataset is built from rich and diverse sources with strong quality control. A large portion of the data was annotated or reviewed by certified TCM practitioners, ensuring both authority and practical value.

The evaluation approach is well-designed. The proposed Ladder-Score better captures terminology accuracy and semantic alignment in TCM question answering, offering more domain-specific reliability than traditional metrics like BLEU and ROUGE.

---

> ### Author Rebuttal · Authors · 2025-07-31
>
> We sincerely thank you for your thorough review and insightful feedback. We greatly appreciate your recognition of the contributions made by *TCM-Ladder* in addressing the absence of standardized multimodal benchmarks in the TCM domain. Your valuable suggestions have further encouraged us to refine and enhance the benchmark to better serve the research community.
>
> In the following, we provide point-by-point responses to each of your comments and suggestions.
>
>
> **1\. Reviewer comment:**
>
> > *The benchmark primarily focuses on closed-ended question answering, with relatively few generative or open-ended QA tasks.*
>
> **Response:**
>
> Thank you for your thoughtful observation. We acknowledge that the current version of the benchmark predominantly focuses on closed-ended question answering formats. This design choice was made with the intention of enabling objective and scalable evaluation across a broad spectrum of subfields and modalities in the early phase of benchmark development.
>
> We fully agree that incorporating more generative and open-ended QA tasks is crucial for assessing the reasoning, explanation, and generative capabilities of large language models in the TCM domain.  In response to this concern, we have integrated such tasks into the benchmark platform. Specifically, we have integrated representative datasets such as Huatuo-26M, TCMD, and CMD, which contain free-form diagnostic reasoning and dialogue-based question answering. These additions reflect our initial effort to address the need for more diverse evaluation formats.
>
> We plan to significantly expand this category in future versions to support rigorous and nuanced model assessment. We sincerely appreciate your suggestion and will continue refining *TCM-Ladder* to ensure a more balanced and representative evaluation framework.
>
>
> **2\. Reviewer comment:**
>
> > *Although the dataset includes audio and video modalities, the current evaluation tasks are primarily focused on text and image. The potential of modalities such as pulse diagnosis audio and acupuncture videos remains underexplored.*
>
> **Response:**
>
> Thank you for your valuable comment. We agree that the current evaluation primarily centers on textual and visual modalities, and we appreciate your suggestion to further explore the potential of audio and video data.
>
> In future work, we plan to develop dedicated benchmark tasks that make fuller use of pulse diagnosis audio and acupuncture videos, such as training multimodal diagnostic models and developing assistive learning tools for acupuncture education. These modalities are integral to real-world TCM practice, and our dataset provides a foundation for such applications. These multimodal resources will also serve as a valuable foundation for future research by other scholars in the field. We believe these additions will further enhance the practical value and research potential of *TCM-Ladder*.
>
> **3\. Reviewer comment:**
>
> > *While the proposed metric is conceptually sound, it would be strengthened by ablation studies or alignment analysis with human evaluations to validate the individual contributions of TermScore and SemanticScore.*
>
> **Response:**
>
> Thank you very much for the valuable suggestion. We agree that further validation of the proposed *LadderScore* would enhance its credibility. While we have provided case studies in the appendix to illustrate how *TermScore* and *SemanticScore* jointly influence the final score, we acknowledge the importance of more rigorous quantitative validation.
>
> In future work, we plan to conduct ablation studies by isolating each component (i.e., using only *TermScore* or only *SemanticScore*) to assess their individual correlation with human judgment. Additionally, we will perform alignment analyses to evaluate how well each component aligns with expert evaluation across different TCM subfields. These studies will help clarify the relative contributions of the two components and guide potential refinements of the metric.

---

> > ### Comment · Reviewer_rBn6 · 2025-08-04
> >
> > Thank you for your response. You have provided a thoughtful and forward-looking plan for future work. Although your replies did not fully resolve the issues that were raised, I believe your work fills a meaningful gap in the field. Therefore, I remain inclined to give this submission a positive evaluation.

---

### Official Review · Reviewer_xDMW · 2025-07-01

**Ethics Flags:** Data privacy, copyright, and consent
**Rating:** 6
**Confidence:** 5

**Summary:**

This submission introduces TCM-Ladder, the first large-scale multimodal question-answering (QA) benchmark specifically designed for evaluating large language models (LLMs) in Traditional Chinese Medicine (TCM). The collected dataset spans multiple TCM subfields (e.g., fundamental theory, diagnostics, herbal formulas, etc.) and integrates diverse modalities (text, images, audio, videos), totaling over 52,000 QA instances. Key contributions include:

1. A comprehensive and large-scale multimodal dataset to address the lack of standardized TCM evaluation frameworks. This work is robust and foundational, providing a solid and comprehensive multimodal foundation for advancing TCM-specific large language model evaluation.

2. Six task types to assess model capabilities: single-choice questions; multiple-choice questions; fill-in-the-blank; long-form diagnostic question answering (dialogue); image-based comprehension tasks; and additional audio and video tasks. These tasks cover a wide range of capabilities such as knowledge recognition, comprehensive reasoning, clinical diagnosis, generative accuracy, and multimodal understanding, providing systematic and multimodal effective tools for the comprehensive evaluation of TCM (Traditional Chinese Medicine) domain models' overall performance.

3. Ladder-Score, an evaluation metric combining TCM terminology accuracy and semantic alignment. This offers a nuanced and rigorous evaluation metric that enhances the precision and domain-specific relevance of assessing TCM model performance.

4. Benchmarking 14 LLMs (9 general-domain, 5 TCM-specific, including 2 proposed by the authors) to highlight domain-specific model advantages.

5. An interactive website for data exploration.

**Dataset Code Accessibility:**

Yes

**Dataset Code Comments:**

The dataset is hosted on Huggingface Hub.

**Ethical Comments:**

The authors stated in the appendix (contained in the supplementary material) `J. Limitations and societal impact` that

```
Visual data such as tongue and herb images raise privacy concerns and require informed consent and anonymization.
```

The question is whether the authors have obtained informed consent from the patients whose tongue images are included in the dataset. If not, this could be a potential ethical issue. The authors should clarify this point in the paper (or in the appendix).

**Ethical Considerations:**

Yes, there are ethics concerns that require attention by the authors

**Final Justification:**

### Issues Resolved

All concerns were adequately addressed: the authors committed to improving figure readability, clarified the seniority of human evaluators with plans to expand specialist annotators, resolved terminology inconsistencies, and confirmed ethics compliance.

### Issues Unresolved

None.

### Reason for Score 6 (Strong Accept)

This work establishes the first large-scale, multimodal benchmark for evaluating TCM-specific LLMs. It demonstrates extensive effort through a high-quality multimodal dataset, rigorous design, and comprehensive benchmarking. Besides, the well-written manuscript further reinforces its readiness for acceptance.

**Limitations Weaknesses:**

This work is solid, I can only raise two possible limitation:

1. In section 5.3, it is stated that the human evaluation involved only 2 TCM physicians (senior or junior?). This has the risk of bias due to its small scale. The reported human accuracy (64%), 4% lower than Bencao, may not reflect true clinical expertise. Suggestion: Include more evaluators (e.g. 5–10) with diverse specialties to improve reliability.

2. Some inconsistencies in the text: line 131-132:

```
... spans a wide range of TCM subfields, including basic theory, diagnostics, internal medicine, surgery, pediatrics, and pharmacology.
```

line 144-145:

```
... we identified seven subfields: fundamental theory, diagnostics, **herbal formulas**, internal medicine, surgery, pharmacognosy, and pediatrics.
```

**Strengths Contributions:**

1. TCM-Ladder is the first multimodal and large-scale TCM benchmark, largely filling gaps and limitations of prior datasets. Its integration of visual, auditory, and video data aligns with TCM’s clinical practice, making evaluations more realistic.

2. The proposed dataset covers 7 TCM subfields and 6 task types, enabling holistic model assessment. The assessment of 14 LLMs demonstrates the benchmark's robustness and relevance, providing a comprehensive evaluation of both general-domain and TCM-specific models.

3. The paper is well-organized, with logical flow from problem motivation to dataset construction, evaluation, and results. Figures and tables effectively summarize key details. A minor suggestion for improvement: slightly enlarge the figures to page width for better visibility. Currently, the font size in most figures is small, making it a little hard to read.

---

> ### Author Rebuttal · Authors · 2025-07-31
>
> Thank you for your thoughtful and comprehensive review of our work. We truly appreciate your recognition of the novelty and significance of *TCM-Ladder* as the first large-scale multimodal benchmark specifically designed for evaluating LLMs in the TCM domain. Your encouragement further motivates us to refine the benchmark and expand its utility for the research community.
>
> In the following, we provide point-by-point responses to your comments below, detailing our clarifications and planned improvements to *TCM-Ladder*.
>
>
> **1\. Reviewer comment:**
>
> > *A minor suggestion for improvement: slightly enlarge the figures to page width for better visibility. Currently, the font size in most figures is small, making it a little hard to read.*
>
> **Response:**
>
> Thank you for the helpful suggestion. We agree that some figures currently have small font sizes that affect readability. We will adjust the font sizes and enlarge the figures in the final version to ensure better visibility for readers. We appreciate your attention to presentation quality.
>
> **2\. Reviewer comment:**
>
> > *In section 5.3, it is stated that the human evaluation involved only 2 TCM physicians (senior or junior?). This has the risk of bias due to its small scale. The reported human accuracy (64%), 4% lower than Bencao, may not reflect true clinical expertise.*
>
> **Response:**
>
> Thank you for pointing this out. The current human evaluation was conducted by two senior TCM physicians holding senior titles, equivalent to associate chief physician or above in the Chinese healthcare system. We will also clarify this in the final version.
>
> We fully acknowledge your concern that relying on individual evaluators to assess tasks spanning multiple TCM subfields may introduce a risk of bias.
>
> In fact, during the evaluation process, both physicians emphasized the challenge of maintaining high confidence across all domains. For example, although they are highly knowledgeable about the pharmacological properties and clinical applications of Chinese herbal medicines, they encountered difficulties when asked to identify herbs solely based on images. The challenge became especially evident when the herbs appeared in different visual forms, such as raw botanical specimens, dried slices, or moist decoctions, which often vary significantly in appearance. According to their feedback, such recognition tasks, especially those involving distinctions among various processed forms of herbs, are better handled by trained dispensary pharmacists than by clinical practitioners. We will incorporate this clarification into the final version of the paper to provide better context for the human evaluation results.
>
> **3\. Reviewer comment:**
>
> > *Suggestion: Include more evaluators (e.g. 5–10) with diverse specialties to improve reliability.*
>
> **Response:**
>
> Thank you for the valuable suggestion. We fully agree that involving more evaluators with diverse TCM specialties can enhance the reliability and comprehensiveness of the human evaluation. In fact, we have already planned to expand our pool of annotators in future iterations of this work. Specifically, we intend to include around 10 experienced TCM experts covering various subfields such as internal medicine, dermatology, acupuncture, and herbal pharmacology. This will allow us to better capture domain-specific nuances and further validate the evaluation results across different modalities.
>
> **4\. Reviewer comment:**
>
> > *Some inconsistencies in the text: line 131-132: “... spans a wide range of TCM subfields, including basic theory, diagnostics, internal medicine, surgery, pediatrics, and pharmacology.”
> > line 144-145: “... we identified seven subfields: fundamental theory, diagnostics, herbal formulas, internal medicine, surgery, pharmacognosy, and pediatrics”*
>
> **Response:**
>
> Thank you for noticing this inconsistency. We acknowledge that the list of TCM subfields in lines 131–132 ("basic theory, diagnostics, internal medicine, surgery, pediatrics, and pharmacology") is not fully aligned with the more comprehensive list presented in lines 144–145, where we specify seven distinct subfields: "fundamental theory, diagnostics, herbal formulas, internal medicine, surgery, pharmacognosy, and pediatrics." To ensure clarity and consistency throughout the paper, we will revise the earlier mention to match the category shown in Figure 2 and lines 144–145.
>
> We appreciate your careful review and will ensure this correction is included in the final version.
>
> **5\. Reviewer comment:**
>
> > *Ethical Considerations: Yes, there are ethics concerns that require attention by the authors
> > Ethics Flags: Data privacy, copyright, and consent*
>
> **Response:**
>
> Thank you for pointing out these important ethical considerations. All clinical data included in *TCM-Ladder* were collected under the approval of institutional review boards (IRBs) with appropriate informed consent obtained from all participants. For sensitive modalities such as tongue images, we applied strict anonymization procedures, including cropping identifying features and removing personal identifiers. All third-party data used in the dataset have been carefully verified for copyright compliance.
>
>
> **6\. Reviewer comment:**
>
> > *The authors stated in the appendix (contained in the supplementary material) J. Limitations and societal impact that Visual data such as tongue and herb images raise privacy concerns and require informed consent and anonymization.
> > The question is whether the authors have obtained informed consent from the patients whose tongue images are included in the dataset. If not, this could be a potential ethical issue. The authors should clarify this point in the paper (or in the appendix).*
>
> **Response:**
>
> Thank you for your thoughtful comment regarding data ethics. We would like to clarify that all visual data used in our benchmark, including tongue and herb images, were collected with informed consent from participants. For clinical tongue images, the consent process was conducted under the supervision of partnering hospitals and institutions, following approved ethical protocols. All identifiable information has been anonymized prior to release. Specifically, we also applied necessary de-identification processing to tongue images to ensure that no facial features or patient-identifying cues remain. We will make sure to clearly state this in the final version of the appendix.

---

> > ### Comment · Reviewer_xDMW · 2025-08-06
> >
> > Thank you for the comprehensive and thoughtful point-by-point responses. I appreciate the detailed clarifications and commitments to address all raised concerns in the final version (figure readability, evaluator details, subfield consistency). The plans to invite more evaluators are welcome. Regarding ethics concerns, as AC has assigned Ethics Reviewers, I defer to their assessment on this matter. Overall, I find the authors' rebuttal satisfactory and have no further major concerns.

---

### Official Review · Reviewer_nad2 · 2025-07-02

**Rating:** 5
**Confidence:** 3

**Summary:**

This paper introduces TCM-Ladder, a multimodal benchmark for evaluating large language models (LLMs) in the context of Traditional Chinese Medicine (TCM). The benchmark spans text, images, audio, and video and includes over 52,000 questions across seven TCM subfields (e.g., diagnostics, herbal formulas, internal medicine). The dataset supports diverse question types, including multiple choice, fill-in-the-blank, visual QA, and long-form diagnostic dialogues. The authors also propose Ladder-Score, a new evaluation metric incorporating TCM terminology accuracy and semantic alignment. Empirical results demonstrate the advantages of domain-specific LLMs (e.g., Bencao, Ladder-base) over general models on various TCM-specific tasks. The dataset, code, and leaderboard are publicly released.

**Dataset Code Accessibility:**

Yes

**Ethical Considerations:**

No, there are no or only very minor ethics concerns

**Final Justification:**

The rebuttal has addressed all my concerns. I will keep my score.

**Limitations Weaknesses:**

This is a well-written and well-executed paper. The authors make a clear and valuable contribution by releasing the first multimodal benchmark tailored for Traditional Chinese Medicine (TCM). The dataset is thoughtfully designed, and the experiments are comprehensive across various LLMs.

My main concern, however, lies in the potential impact and resonance within the NeurIPS community. Since TCM is primarily practiced and studied within Chinese-speaking regions, and most of the content in the dataset is in Chinese, it may be challenging for international researchers to adopt or build upon this work without sufficient domain or language expertise. This could limit the broader applicability or visibility of the benchmark in the global machine learning community.

Additionally, while the dataset includes audio and video modalities, the current experiments and evaluations focus almost exclusively on text and images. To fully demonstrate the dataset’s multimodal value, I would encourage the authors to include clearer task definitions or pilot experiments involving these modalities. Finally, although the benchmark is currently designed for LLMs, it would be beneficial to assess whether traditional multimodal models (e.g., visual-text transformers or audio encoders) can effectively operate on this dataset, to broaden its utility beyond LLM-centric paradigms.

**Strengths Contributions:**

1.	This is the first benchmark to comprehensively integrate visual (e.g., tongue/herb images), audio, and video modalities into the evaluation of LLMs in the TCM domain, going far beyond existing purely text-based datasets.
2.	Data is professionally curated—many QA pairs are manually written or verified by licensed TCM physicians. Ethical approval for sensitive data (e.g., tongue images) is also addressed.
3.	The dataset is well-structured across six task types, covering both structured knowledge assessment (e.g., MCQ) and clinical reasoning (e.g., long-form dialogue), which mirrors the diversity and complexity of real-world TCM practice.
4.	Novel and practical evaluation metric (Ladder-Score): Combines domain-specific term matching and LLM-based semantic scoring to better assess answer quality in TCM-specific QA, addressing known limitations of BLEU/ROUGE in this context.
5.	The authors benchmark 14 LLMs (9 general, 5 TCM-specific) and show meaningful performance gaps.
6.	Dataset, leaderboard, and model code are publicly released, with a live platform to support continued benchmark expansion and model tracking.

---

> ### Author Rebuttal · Authors · 2025-07-31
>
> Thank you sincerely for your thoughtful summary and for your recognition of the contributions presented in our work. We greatly appreciate your positive evaluation of the novelty and utility of the *TCM-Ladder* benchmark. Your comments further motivate us to continuously improve *TCM-Ladder* as a comprehensive and multimodal benchmark for Traditional Chinese Medicine.
>
> Below, we provide point-by-point responses to any specific comments or suggestions to further improve our work.
>
>
> **1\. Reviewer comment:**
>
> > This is a well-written and well-executed paper. The authors make a clear and valuable contribution by releasing the first multimodal benchmark tailored for Traditional Chinese Medicine (TCM). The dataset is thoughtfully designed, and the experiments are comprehensive across various LLMs.
> >
> > My main concern, however, lies in the potential impact and resonance within the NeurIPS community. Since TCM is primarily practiced and studied within Chinese-speaking regions, and most of the content in the dataset is in Chinese, it may be challenging for international researchers to adopt or build upon this work without sufficient domain or language expertise. This could limit the broader applicability or visibility of the benchmark in the global machine learning community.
>
> **Response:**
>
> Thank you very much for your thoughtful and encouraging review. We sincerely appreciate your recognition of the value and execution of the *TCM-Ladder* benchmark.
>
> We fully agree that the Chinese language and the domain-specific nature of TCM pose challenges for broader adoption. We are currently working on translating the corpus and task definitions into English to reduce the language barrier. However, this process is highly time-consuming and labor-intensive, as many samples involve Classical Chinese and technical terminology that are difficult for current translation models to accurately translate into English or other languages. Expert manual translation is required to preserve semantic accuracy. We are committed to releasing an English-accessible version of the benchmark as soon as possible.
>
> **2\. Reviewer comment:**
>
> > *Additionally, while the dataset includes audio and video modalities, the current experiments and evaluations focus almost exclusively on text and images. To fully demonstrate the dataset’s multimodal value, I would encourage the authors to include clearer task definitions or pilot experiments involving these modalities.*
>
> **Response:**
>
> Thank you for your comments. In the final version of the paper, we will further emphasize the multimodal value of the dataset. Specifically, *TCM-Ladder* can support the development of domain-specific multimodal models in TCM, where such models are currently scarce. This is particularly relevant because the fundamental diagnostic process in TCM involves multiple sensory modalities, including visual inspection, auditory cues, patient inquiry, and tactile information. The inclusion of audio, visual, and other modalities in the dataset provides a valuable benchmark for training and evaluating future multimodal models tailored to the TCM domain.
>
> **3\. Reviewer comment:**
>
> > *Finally, although the benchmark is currently designed for LLMs, it would be beneficial to assess whether traditional multimodal models (e.g., visual-text transformers or audio encoders) can effectively operate on this dataset, to broaden its utility beyond LLM-centric paradigms.*
>
> **Response:**
>
> Thank you for this valuable suggestion. We fully agree that assessing traditional multimodal models (e.g., visual-text transformers, audio encoders) on the *TCM-Ladder* benchmark would broaden its applicability beyond LLM-centric paradigms.
>
> As a next step, we plan to release baseline performance results of non-LLM multimodal models on relevant *TCM-Ladder* tasks to help position the benchmark within the broader multimodal AI landscape. We believe this direction is essential to encourage community contributions from various research domains, including multimodal fusion, domain adaptation, and medical vision-language learning.

---

> > ### Comment · Reviewer_nad2 · 2025-08-06
> >
> > Thank you for the classification, which addressed my concerns. I will maintain my scores.

---

### Official Review · Reviewer_Mg3o · 2025-07-03

**Rating:** 5
**Confidence:** 3

**Summary:**

The paper proposes TCM-Ladder - a large multi-modal benchmarking dataset for Traditional Chinese Medicine (TCM), consisting of 52k+ questions spanning single and multiple-choice, fill-in-the-blank, diagnostic dialogue and visual comprehension tasks. This is the first multi-modal dataset of its kind, addressing the growing need for evaluation on LLMs on TCM tasks and building on top of previous work by including multiple modalities.

The paper evaluates a number of general domain and task-specific models and proposes a GRPO-trained model that achieves state-of-the-art performance on the benchmark. Moreover, the paper introduces a new metric - LadderScore - designed specifically for evaluation TCM QA by integrating both terminology usage and semantic expression of answers. The multi-modal benchmark dataset is available for exploration online and welcomes submissions from the community.

**Dataset Code Accessibility:**

Yes

**Dataset Code Comments:**

Dataset is well documented on the webpage at https://github.com/orangeshushu/TCM-Ladder.
I am able to visualize the dataset on Huggingface, but am not able to load it with the code snipped. The HuggingFace Dataset Card is missing information

**Ethical Comments:**

The paper clearly states that it all data collection procedures were approved by the institutional ethics review board at top of page 5.

**Ethical Considerations:**

No, there are no or only very minor ethics concerns

**Limitations Weaknesses:**

**Metrics**
- Human Evaluation seems to be around 64% for top-1 accuracy for answer retrieval, with the best performing LLM surpassing that. This is explained in the paper as being due to the challenge of finding human experts that are knowledgeable in all the multiple modalities and subfields. I found this a bit confusing, as it would mean that an expert TCM practitioner would only be able to obtain around 64% accuracy on this data, which would make the benchmark either too challenging, or perhaps containing information that is outside of the distribution of knowledge for a TCM practitioner.

- Moreover, if finding interdisciplinary expertise is of concern, have the authors considered on performing expert human evaluation for each individual subfield, and aggregating the scores? The human evaluation score would likely go up.

**Evaluation Metric**: It's a bit unclear how the choice of the Ladder-Score was determined and how alpha, beta were chosen in Ladder-Score = alpha * TermScore + beta * SemanticScore

**New submissions**
- How will new submissions from the community be integrated into the dataset? What quality control checks will be applied to ensure quality is preserved?

**Minor**
- The URL in the abstract gives an error - unclear why it's needed if the tcmladder.com is provided

**Strengths Contributions:**

**Utility**: The size of the dataset is pretty high, with more than 52k samples, and spans multiple tasks and modalities, being a rich source for multi-modal data that can be used for both training and evaluation of multi-modal models.

**Dataset Quality**: Dataset Creation seems to follow a high-quality standard - the dataset creation methodology is described in detail, domain-experts (licensed-physicians) have been consulted and involved in the process, and proper consideration has been given to human subjects through institutional review boards and mitigation through data handling (e.g. cropping of the tongue images to appropriate patches). This ensures that the data has been ethically obtained, post-processed properly and validated by domain experts.

**Dataset Presention**: The webpage provides a clear presentation of the data and is easily accessible; allowing submissions from the community is beneficial, as long as proper guardrails are in place to ensure quality.

---

> ### Author Rebuttal · Authors · 2025-07-31
>
> We sincerely thank you for your positive and thoughtful overall evaluation of our work. We truly appreciate your recognition of the novelty and utility of the *TCM-Ladder* benchmark. Your encouraging comments are greatly appreciated and provide strong motivation for us to continue improving and extending this resource.
>
> In the following, we provide point-by-point responses to each of your comments and suggestions.
>
> **1\.Reviewer comment:**
>
> > *Human Evaluation seems to be around 64% for top-1 accuracy for answer retrieval, with the best performing LLM surpassing that. This is explained in the paper as being due to the challenge of finding human experts that are knowledgeable in all the multiple modalities and subfields. I found this a bit confusing, as it would mean that an expert TCM practitioner would only be able to obtain around 64% accuracy on this data, which would make the benchmark either too challenging, or perhaps containing information that is outside of the distribution of knowledge for a TCM practitioner.*
>
> **Response:**
>
> Thank you for your insightful comment. We acknowledge that the observed human evaluation accuracy of approximately 64% for top-1 answer retrieval may appear unexpectedly low. However, we would like to clarify that this figure should not be interpreted as an upper bound of expert performance in TCM. Instead, it reflects the inherent limitations of employing only two senior TCM practitioners to assess a benchmark encompassing a wide range of highly specialized subdomains such as herbal image recognition, acupuncture techniques, pulse diagnosis, and classical text interpretation.
>
> Both evaluators expressed challenges in maintaining consistently high confidence across all modalities and tasks, particularly in domains outside their primary areas of expertise. For example, they noted difficulty in identifying the same herb presented in varying visual forms (e.g., raw botanical specimens, processed decoctions, or dried slices), a task typically handled by trained dispensary pharmacists rather than clinical practitioners.
>
> Furthermore, the multi-modal and interdisciplinary nature of the benchmark exceeds the typical cognitive scope of any individual TCM expert. Therefore, the relatively moderate human performance reflects the breadth and depth of the dataset rather than its unsuitability or excessive difficulty.
>
> **2\. Reviewer comment:**
>
> > *Moreover, if finding interdisciplinary expertise is of concern, have the authors considered on performing expert human evaluation for each individual subfield, and aggregating the scores? The human evaluation score would likely go up.*
>
> **Response:**
>
> Thank you for the helpful suggestion. We fully agree that performing expert evaluations within individual subfields and aggregating the results could lead to a more accurate reflection of human performance. In fact, this aligns with our future evaluation plan. Due to resource limitations, the current version of the benchmark relied on only two senior TCM physicians to conduct holistic evaluations across all tasks. However, as you rightly pointed out, the interdisciplinary nature of the dataset poses challenges for any single evaluator to perform consistently well across all modalities and domains.
>
> To address this, we plan to recruit approximately 10 senior TCM practitioners, each with expertise in different subfields such as herbal pharmacology, acupuncture, internal medicine, and classical TCM texts, to independently evaluate the subset of tasks relevant to their specialty. We will then aggregate their evaluations to produce a more comprehensive and representative human performance score. We believe this approach will not only improve reliability but also provide a fairer and granular assessment of LLM performance in specialized areas of Traditional Chinese Medicine.
>
> **3\. Reviewer comment:**
>
> > *Evaluation Metric: It's a bit unclear how the choice of the Ladder-Score was determined and how alpha, beta were chosen in Ladder-Score = alpha \* TermScore + beta \* SemanticScore*
>
> **Response:**
>
> Thank you for the thoughtful comment. The choice of the *Ladder-Score* formulation was based on both empirical validation and task-specific design considerations. Specifically, we aimed to balance two complementary dimensions in evaluating TCM answers: (1) factual correctness and domain adherence (captured by *TermScore*, based on a curated TCM terminology dictionary), and (2) semantic fluency and clinical reasoning coherence (captured by *SemanticScore*, computed using GPT-4 with rubric-based prompting).
>
> We conducted a grid search on a validation set with human expert annotations to explore different weight combinations. The best alignment with expert ratings was obtained when α = 0.6 and β = 0.4, indicating a moderate preference for factual and terminological accuracy while still valuing semantic quality.
>
> In future work, we plan to explore adaptive weighting schemes, for example, adjusting α and β based on task type (e.g., short vs. long-form responses) or difficulty level.
>
> **4\. Reviewer comment:**
>
> > *How will new submissions from the community be integrated into the dataset? What quality control checks will be applied to ensure quality is preserved?*
>
> **Response:**
>
> Thank you for raising this important point. We have designed *TCM-Ladder* as a living benchmark that encourages continued community contributions. To ensure both scalability and quality, we have planned the following integration and quality control pipeline for new submissions:
>
> 1. Submission format and metadata requirements\
>    All community-submitted samples must follow a standardized format, including task type, input-output specification, answer key, and metadata annotations (e.g., difficulty level, source, modality). A submission template will be provided.
> 2. Manual review by domain experts\
>   Each submission will be reviewed by domain experts to ensure accuracy and quality. Submissions with ambiguous, incorrect, or non-standard content will be filtered out.
> 3. Cross validation and redundancy checks:
>    For tasks involving subjective or complex labels, at least two independent annotators will assess the consistency. Only samples passing a minimum agreement threshold will be retained. We will apply automatic checks for semantic duplication.
> 4. Community feedback\
>    We have added a "Submission & Feedback" page to the *TCM-Ladder* website. The community can report suspicious entries and propose concrete revisions. These flagged samples will be reviewed by licensed TCM experts, and validated changes will be incorporated into periodic versioned updates.
>
> We believe this framework balances openness and robustness, ensuring that *TCM-Ladder* continues to grow without compromising quality.
>
> **5\. Reviewer comment:**
>
> > *The URL in the abstract gives an error - unclear why it's needed if the tcmladder.com is provided*
>
> **Response:**
>
> Thank you for your comment. Before the official launch of our website, we conducted internal accessibility tests and found that certain university networks may block non-whitelisted domain names. To ensure maximum accessibility for users and reviewers during the evaluation phase, we temporarily included an alternative server IP address in the abstract as a backup access point. If this is no longer necessary, we will remove the redundant link in the final version.
>
>
> **6\. Reviewer comment:**
>
> > *I am able to visualize the dataset on Huggingface, but am not able to load it with the code snipped. The HuggingFace Dataset Card is missing information.*
>
> **Response:**
>
> Thank you for pointing this out. Due to restrictions during the review phase, we are currently unable to modify the HuggingFace dataset page. We have revised our dataset card configuration and will update it after the review period. You may download the specific Parquet file and load it directly for now. We greatly appreciate your valuable feedback and continued support.

---

### Decision · Program_Chairs · 2025-09-18

**Decision:**

Accept (poster)

**Comment:**

This paper introduces TCM-Ladder, the first large-scale, multimodal benchmark for Traditional Chinese Medicine, spanning text, image, audio, and video modalities with 52K+ diverse QA instances. The dataset is well-curated with expert involvement and thoughtful task design, and the proposed Ladder-Score offers a domain-aware evaluation metric.

Reviewers appreciated the dataset’s breadth, structure, and relevance, particularly its potential to advance research in underrepresented domains. Nonetheless, some concerns were raised regarding the limited utilization of audio/video modalities, the clarity of metric components, and the relatively narrow linguistic scope. While these aspects could be further strengthened, the submission is technically solid and addresses an important gap. I recommend a borderline accept.